# Report on Landsat 8 and Sentinel-2B observations of the Nord Stream 2 pipeline methane leak

Matthieu Dogniaux[1], Joannes D. Maasakkers[1], Daniel J. Varon[2], and Ilse Aben[1]

[1]SRON Netherlands Institute for Space Research, Leiden, The Netherlands
[2]School of Engineering and Applied Science, Harvard University, Cambridge, 02138, USA

**Correspondence:** Matthieu Dogniaux (M.Dogniaux@sron.nl)

**Abstract.** In late September 2022, explosions of the Nord Stream pipelines caused what could be the largest anthropogenic methane leak ever recorded. We report on Landsat 8 (L8) and Sentinel-2B (S-2B) observations of the sea foam patch produced by the Nord Stream 2 (NS2) leak located close to Bornholm Island, acquired on September 29[th] and 30[th], respectively. Usually, reflected sunlight over sea is insufficient for these Earth-imagers to observe any methane signal in nadir-viewing geometry. However, the NS2 foam patch observed here is bright enough to possibly allow the detection of methane above it. We apply the
Multi-Band Single-Pass (MBSP) method to infer methane enhancement above the NS2 foam patch and then use the Integrated Mass Enhancement (IME) method in a Monte Carlo ensemble approach to estimate methane leak rates and their uncertainties. This very specific NS2 observation case challenges some of MBSP and IME implicit assumptions, and thus calls for customized calibrations: (1) for MBSP, we perform an empirical calibration of sea foam albedo spectral dependence by using sea foam observations in ship trails, and (2) for IME, we yield a tailored effective wind speed calibration that accounts for a partial
plume observation, as methane enhancement may only be seen above the NS2 sea foam patch. Our comprehensive uncertainty analysis yields large methane leak rate uncertainty ranges that include zero, with a best estimate of $501\pm521$ t/hr. Thus, no firm conclusion can be drawn from the single or combined overpasses of L8 and S-2B. Within all our Monte Carlo ensembles, positive methane leak rates have higher probabilities ($79 - 88\%$) than negative ones ($12 - 21\%$), thus indicating that L8 and S-2B likely captured a methane-related signal. Overall, we see our work both as a nuanced analysis of L8 and S-2B
contributions to quantifying the NS2 leak emissions and as a methodological cautionary tale that builds insight on MBSP and IME sensitivities.

## 1 Introduction

From September 26[th] to October 2[nd], 2022, leaks occurred on the Nord Stream (NS) and Nord Stream 2 (NS2) pipelines in the Baltic Sea. They caused intensive bubbling and extensive foam patches at the sea surface, as well as methane emissions that
could be one of the strongest methane leak events ever recorded (Sanderson, 2022). The Southern NS2 sea foam patch close to Bornholm Island was observed on September 29[th] and 30[th] by Landsat 8 and Sentinel-2B (respectively), two Earth-imaging satellites that are sensitive to large methane point sources (Varon et al., 2021). We report on those two observations, and exhibit the challenges they come with to evaluate the NS2 methane leak rate.

Anthropogenic methane emissions are the second largest contributor to human-induced climate change, and their drastic reduction is required to keep global warming below 1.5°C or 2.0°C (IPCC, 2021). In the past decade, developments in space-based methane observation have had a transformative impact on methane super-emitter detection and monitoring, and can contribute to track progress towards the Paris Agreement goals (e.g. Nisbet et al., 2020). Among them, the TROPOspheric Monitoring Instrument (TROPOMI, Veefkind et al., 2012; Lorente et al., 2021) measures back-scattered sunlight in the short-wave infrared (SWIR) around 2.3 $\mu$m at 0.25 nm resolution, at a moderate $5.5 \times 7$ km$^2$ spatial resolution at nadir and with daily global coverage. Global methane concentrations maps are drawn from these measurements using a full-physics approach which accounts for geophysical variables besides methane (e.g. albedo, water vapour, aerosol optical depth, etc) that could interfere in the retrieval process (Lorente et al., 2021). Its observations have been successfully used to detect and estimate anthropogenic methane emissions arising from various point or localized sources (e.g. Pandey et al., 2019; Lauvaux et al., 2022; Schuit et al., 2023). SWIR satellite instruments with higher spatial resolution (few tens of meters) have proved complementary by enabling the identification of methane emission sources at facility-scale. These notably include the methane-dedicated GHGSat constellation (Jervis et al., 2021) and Earth-imagers such as Sentinel-2 or Landsat 8. Earth-imagers are not spectrally resolved like TROPOMI and were not originally designed to measure greenhouse gases. However, under the right conditions (bright, quasi-homogeneous land surface), their methane sensitive bands ($\sim$ 100-200 nm in width) can be repurposed to retrieve large methane concentration enhancements and image point source emission plumes (e.g. Varon et al., 2021; Irakulis-Loitxate et al., 2022b). Like any other SWIR instrument, these Earth-imagers do not typically offer coverage over water bodies, because the water albedo is too dark at nadir pointing. However, sun-glint observations over sea can allow methane plume detection with these satellites as well (Irakulis-Loitxate et al., 2022a).

When the NS and NS2 leaks occurred, and in the following week, TROPOMI was not able to acquire exploitable data over land in the Baltic Sea vicinity due to cloudiness. However, thanks to their finer spatial resolution, Landsat 8 (L8) and Sentinel-2B (S-2B) have been able to perform nadir-pointing observations showing the Southern NS2 leak on September 29[th] and 30[th], respectively. They did not benefit from sun glint, but the bright foam patch produced by the bubbling leak at the sea surface reflected enough sunlight to consider using the observations, and assess whether a methane signal can be sensed. Besides L8 and S-2B, GHGSat could point their instruments towards the same NS2 leak on September 30[th] and observe a methane emission plume in glint geometry twice (GHGSat, 2022). After initial Twitter reports by the International Methane Emissions Observatory (IMEO, 2022), Jia et al. (2022) published results for the Sentinel-2B observation, acknowledging significant uncertainties in their methodology regarding the spectral reflectance of bubbles and the partial imaging of the methane plume.

This work first aims to show how Landsat 8 and Sentinel-2B observations of the Nord Stream 2 leak challenge implicit assumptions in methods usually applied for Earth-imager methane plume analysis and emission rate quantification. It then proposes to account for identified issues by using customized calibrations, and to assess the possibility of using Landsat 8 and Sentinel-2B to sense and quantify methane emissions from the Nord Stream 2 leak.

This paper is structured as follows: Section 2 describes general aspects about the materials and methods used in this work as well as Nord Stream 2 specific calibrations. Section 3 presents the obtained methane leak rates. Finally, Section 4 highlights the conclusions of this work.

## 2 Materials and methods

This section describes general aspects of the data and methods used here, as well as the custom calibrations that are necessary to adapt them to this singular Nord Stream 2 observation case.

### 2.1 Landsat 8 and Sentinel-2B satellite observations

#### 2.1.1 General aspects

Landsat 8 (hereafter L8) is an Earth-imaging satellite with a swath of 185 km and a revisit time of 16 days. It measures reflected
sunlight over 10 different spectral bands located in the visible, short-wave infrared (SWIR) and thermal infrared, with spatial resolutions ranging from 15 to 100 m (Roy et al., 2014).

The Copernicus Sentinel-2 mission comprises two Earth-imaging satellites (Sentinel-2A and Sentinel-2B, hereafter S-2B) with a swath of 290 km and revisit time of 10 days each, and aims to monitor changes on our Earth's surface. They measure reflected sunlight over 12 different spectral bands located in the visible and SWIR, with spatial resolutions ranging from 10 to
80 60 m (Drusch et al., 2012).

Here, we use Top Of the Atmosphere (TOA) reflectance data observed by L8 and S-2B for two methane sensitive SWIR spectral bands around 1.6 $\mu$m (bands 6 and 11 for L8 and S-2B, respectively) and 2.2 $\mu$m (bands 7 and 12 for L8 and S-2B, respectively). These L8 and S-2B SWIR observations have spatial resolutions of 30 and 20 m, respectively.

#### 2.1.2 Nord Stream 2 leak observations

Figure 1 shows L8 and S-2B TOA reflectance observations of the NS2 methane leak (top panels) and exhibits, using simple empirically-determined thresholds (see Supplements), the different pixel types (dark still sea, NS2 leak, cloud) included in the images by comparing $s_1$ and $s_2$ TOA reflectance values (bottom panels). The L8 image acquired on September 29th, 2022, is composed of the bubbling sea foam patch at its center, surrounded by dark still sea and cloud pixels. The S-2B image acquired on September 30th, 2022 is much cleaner and only includes the NS2 bubbling sea foam patch at its center, surrounded by dark
still sea pixels.

### 2.2 Methane enhancement retrieval: the Multi-Band Single-Pass (MBSP) method

We use the Multi-Band Single-Pass (MBSP) method to retrieve local methane column enhancements from Earth imager observations. We first describe MBSP and its standard calibration approach, and then show how this specific NS2 case study calls for a custom calibration.

#### 2.2.1 General description

The TOA reflectance data can be used to retrieve atmospheric methane concentration enhancements with the Multi-Band Single-Pass (MBSP) method, first proposed by Varon et al. (2021). It relies on the relative change in TOA reflectance $\Delta R$

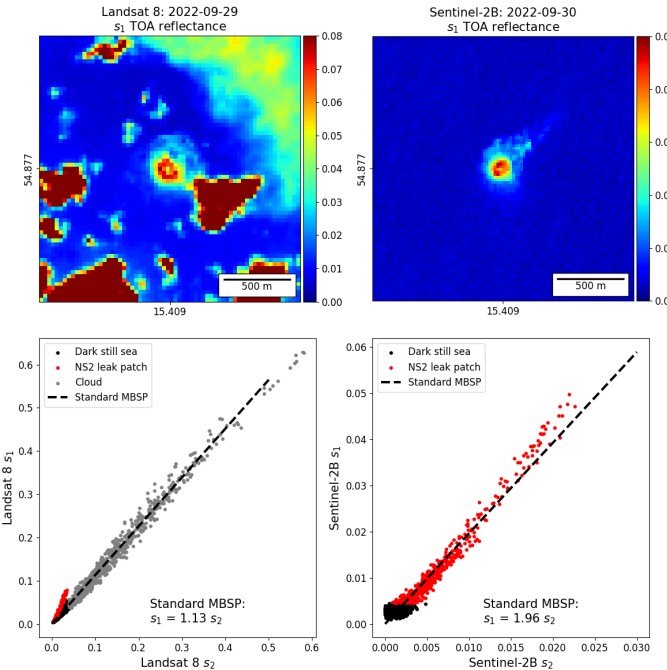

**Figure 1.** Landsat 8 (left, September 29$^{\text{th}}$ 2022) and Sentinel-2B (right, September 30$^{\text{th}}$ 2022) images of the Nord Stream 2 leak for $s_1$ (top), and $s_1$ and $s_2$ TOA reflectance comparisons depicting different pixel natures and showing the standard MBSP $c$ calibration line (bottom). The pixel natures of dark still sea (black), clouds (grey) and NS foam patch (red, all influenced by the methane leak) are separated using empirically determined thresholds given in the Supplements. The standard MBSP calibration (dashed line) is provided here to illustrate why it proves to be unsuitable for this specific NS2 case, as detailed in Sect. 2.2.2.

between two spectral bands $s_1$ (around 1.6 $\mu$m, low sensitivity to methane) and $s_2$ (around 2.2 $\mu$m, strong sensitivity to methane) computed as:

$$\Delta R = \frac{c \times s_2 - s_1}{s_1} \tag{1}$$

with $c$, a linear calibration coefficient fitted on all the pixels included in the target image, to account for any non-methane-related spectral effects between bands $s_1$ and $s_2$, most importantly the spectral dependence of the albedo. This calibration strategy was proposed with the MBSP method by Varon et al. (2021), and implicitly assumes that image-wide pixels are representative of the surface characteristics expected below the (potential) methane plume. Hereafter, we will refer to this "naïve" calibration strategy as the "standard MBSP calibration". The rationale of MBSP is that deviations in the methane-sensitive $s_2$ band from the expected $s_1/s_2$ ratio (captured in the fitted $c$ coefficient) are interpreted as methane enhancements. Pixels with $\Delta R < 0$ relate to higher than expected atmospheric absorption and yield positive methane enhancements. The translation of $\Delta R$ to methane enhancements is performed using pre-computed look-up tables, generated through radiative transfer simulations. Here, they

are based on the 2020 version of the HITRAN spectroscopic database (Gordon et al., 2022), rely on a 21-layer atmospheric model representative of mid-latitudes and include the impact of the solar zenith angle.

### 2.2.2 Empirical calibration of the spectral dependence of sea foam reflectance in MBSP

Here, we seek to determine whether a methane enhancement signal can be retrieved from L8 and S-2B images of the NS2 sea foam patch. No methane signal can be expected to be visible over the dark still sea or the clouds. Consequently, considering the general description of MBSP given in Sect. 2.2.1, properly constraining the spectral dependence of sea foam albedo between $s_1$ and $s_2$ is critical to obtain non-biased methane enhancements through MBSP.

Whitlock et al. (1982) and Koepke (1984) show that we expect a reflectance ratio $s_1/s_2$ over sea foam of about 2 or slightly lower (graphical reading). However, the only pixels representative of sea foam that can be observed in L8 and S-2B images of the NS2 leak are the ones caused by the leak itself, above which we also expect a possible methane enhancement signal. Unlike a land image, it is thus not possible to assess whether the standard MBSP calibration can separate the spectral impact of methane from the spectral dependence of the albedo for this specific NS2 case. This is particularly noticeable in Fig. 1 for the S-2B image, where the standard MBSP calibration is driven by the NS2 sea foam patch ($c = 1.96$). This issue similarly applies to the L8 NS2 observation, that also features an additional complication: very bright clouds are present in the image, which in this case drive the standard MBSP calibration ($c = 1.13$). Thus, the standard MBSP calibration lines included in Fig. 1 illustrate why the NS2 observation case, that relies on a small sea foam patch, calls for an external calibration of the spectral dependence of sea foam albedo.

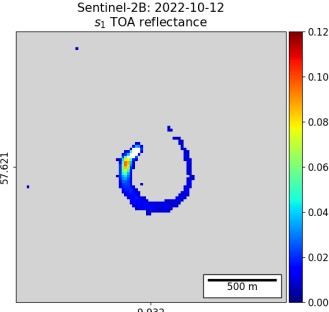

**Figure 2.** Example of sea foam observation in the Sentinel-2B image of a ship trail acquired on October 12[th], 2022. Dark still sea and ship pixels have been removed and are shown in grey and white, respectively. They are also excluded from the sea foam albedo spectral dependence results presented later in Sect. 2.2.2 and in the Supplements.

We therefore empirically constrain the spectral dependence of sea foam albedo by using sea foam observations in ship trails unaffected by methane plumes. We treat each satellite separately in order to account for their different instrumental characteristics. By visual inspection of RGB Sentinel-2 and Landsat data on the EO Browser of Sentinel-Hub (2023), we gather 27 and 38 images of ship trails for L8 and S-2B, respectively, located in the North and Baltic Seas from September and October

2022. For each of these images, we separate ship and sea foam pixels from the dark still sea pixels by using an empirically determined threshold $\tau_1$, such that $s_1 > \tau_1$; and then separate sea foam from ship pixels by applying a second empirically determined threshold $\tau_2$, such that $s_2 < \tau_2$ (Supplement Tables 2 and 3). Figure 2 shows an example of sea foam pixels extracted from an S-2B ship trail image. For each image, using sea foam pixels only, we perform a least-squares linear fit (with an intercept set to zero) of $s_1$ as a function of $s_2$ to determine $c_i$, the coefficient describing the spectral dependence of sea foam

albedo for the $i$-th image (see individual $c_i$ values and fits obtained for each ship trail observation in the Supplements). For L8 and S-2B separately, we then compute $\bar{c}$ as the mean of the individual calibrations. Figure 3 presents the results of this satellite-specific empirical calibration of the spectral dependence of sea foam albedo. We obtain $\bar{c} = 1.96 \pm 0.23$ and $\bar{c} = 1.91 \pm 0.22$ for L8 and S-2B, respectively. These Top-of-the-Atmosphere reflectance ratios are overall consistent with results presented by Whitlock et al. (1982) and Koepke (1984) that were measured on the ground. Comparing the S-2B result to the slightly

higher standard MBSP calibration ($c = 1.96$) also confirms the above mentioned hypothesis that the standard calibration may have captured some methane signal. Indeed, for given fixed $\{s_1, s_2\}$ values, a decrease in the spectral dependence calibration coefficient $c$ (compared to the standard calibration) reduces $\Delta R = (cs_2 - s_1)/s_1$, which translates to an increase of methane enhancement via the use of MBSP.

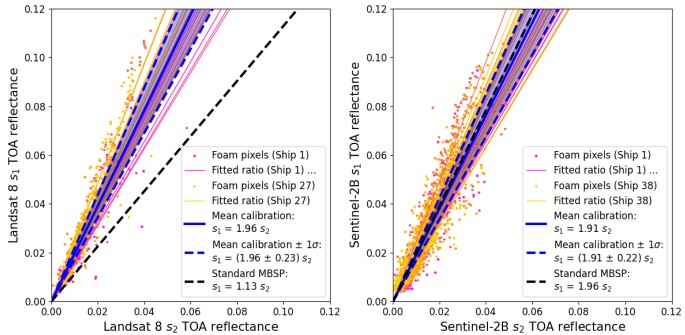

**Figure 3.** Empirically determined sea foam albedo spectral dependence between $s_1$ and $s_2$ for Landsat 8 (left) and Sentinel-2B (right). Sea foam pixels for all ship images are depicted (dots with different colors indicating different ships, the legend only includes elements for the first and last images), along with their respective calibration slopes (thin lines, each is detailed in the Supplements, the legend only includes elements for the first and last images). These enable the computation of the mean and 1-$\sigma$ standard deviation of the empirically determined sea foam albedo spectral dependence (thick full and dashed blue lines). The standard MBSP calibration (thick dashed black line) is also shown.

     MBSP can then be applied using these newly determined empirical calibrations (computing $\Delta R$ using $\bar{c}$). Figure 4 shows

the methane enhancements obtained with the satellite-specific $\bar{c}$ calibration values, and how $s_1$ and $s_2$ TOA reflectance values compare to them. For the L8 observation of the NS2 leak, the sea foam patch pixels show an $s_1/s_2$ ratio of 2.09 (red line), which is slightly higher than the average empirical calibration of the L8 sea foam albedo spectral dependence ($\bar{c} = 1.96 \pm 0.23$), but comprised within its $\pm 1\sigma$ uncertainty interval. This ship-based $\bar{c} - s_1/s_2$ negative difference overall translates to positive methane enhancement through MBSP. On average, we obtain L8 methane enhancement values ranging from -2.5

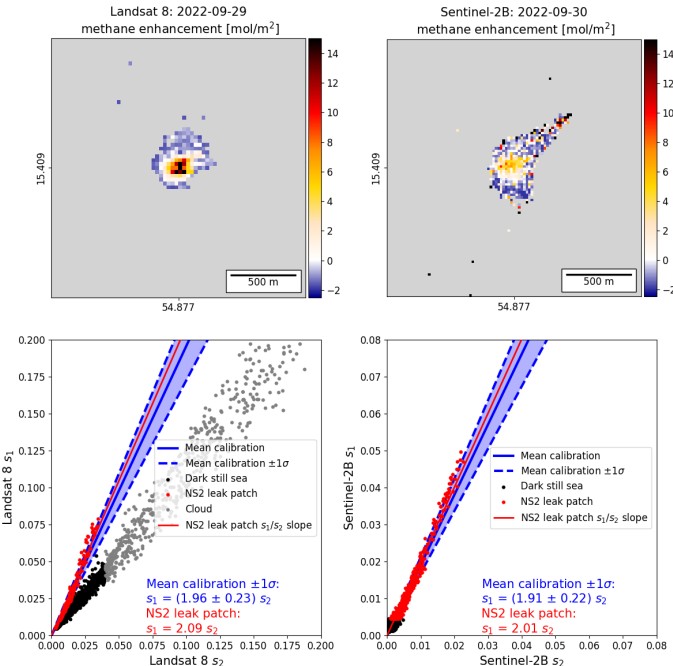

**Figure 4.** Methane enhancement results obtained through MBSP for Landsat 8 (top left, September 29[th] 2022) and Sentinel-2B (top right, September 30[th] 2022), pixels not belonging to the foam patch have been filtered out and shown in grey. Comparisons of $s_1$ and $s_2$ TOA reflectance (bottom) depicting different pixel types and showing the empirically determined spectral dependence of sea foam albedo (thick blue line, the individual ship trail observations underlying this result are shown in Fig. 3 and in supplementary Tables S2 and S3 for L8 and S-2B, respectively), and the $s_1/s_2$ ratio observed over the NS2 sea foam patch (red line). The higher slopes shown by the $s_1/s_2$ ratios (red) compared to the empirical calibrations (blue) are driven by the brightest pixels at the center of the sea foam patch that offer a better signal-to-noise ratio to observe methane absorption than darker pixels.

to 15 mol/m$^2$. Negative enhancements are associated with pixels falling right of the $s_1/s_2$ empirical calibration line (low TOA reflectance values, at the sea foam patch edges), and positive enhancements are associated with pixels falling left of the empirical calibration line (high TOA reflectance values, at the sea foam patch center). The S-2B observation is similar but exhibits more noise, overall showing enhancements from -2.5 mol/m$^2$ on the sea foam patch edges to about 8 mol/m$^2$ at its brighter center.

## 2.3 Emission rate quantification: the Integrated Mass Enhancement (IME) method

We use the Integrated Mass Enhancement (IME) method to quantify the methane emission rate from local methane column enhancement retrievals that show an emission plume. Here, we first explain why we choose the IME method and how it works, then we explain why this specific NS2 case study also calls for a custom calibration for the IME method.

### 2.3.1 General description

If a plume is observed in an image resulting from MBSP, the associated emission rate can be quantified using different approaches such as the Gaussian plume inversion (GP), source pixel (SP), Cross-Section flux (CSF) and Integrated Mass Enhancement (IME) methods (Varon et al., 2018). Because GP and SP are not suited for the quantification of plumes detected using high-resolution satellite observations, and the CSF relies on several transects drawn on an extended downwind plume, we use the IME method. This method was first proposed by Frankenberg et al. (2016) and its calibration and operational use

was improved by Varon et al. (2018). Given a plume, the IME method relates the emission rate $Q$ to the plume's total methane mass and its residence time in the atmosphere. We have:

$$Q = \frac{U_{\text{eff}}}{L} \sum_i \Delta X_{\text{CH}_4 i} \times a_i \tag{2}$$

with $U_{\text{eff}}$, the effective wind speed transporting the plume, $L = \sqrt{\sum_i a_i}$ the plume extent, $X_{\text{CH}_4 i}$, the total column methane enhancement of the $i$-th plume pixel, and $a_i$, the area of this pixel.

Plume transport includes complicated three-dimensional and turbulent effects that require computer-intensive simulations to be accounted for, if even possible given the randomness of turbulence. Through IME, the overall impacts of those effects are presumably captured into a single effective wind speed, denoted $U_{\text{eff}}$. $U_{\text{eff}}$ is calibrated against the 10-m wind speed provided by meteorological models ($U_{10\text{m}}$) over a set of Large Eddy Simulations (LES) made for known synthetic emission rates, and re-sampled according to a given instrument characteristics (spatial resolution, noise model, etc.). Thus, $U_{\text{eff}}$ can be calibrated

for specific instruments and observing conditions. Varon et al. (2021) provide an effective wind speed calibration model for Sentinel-2-like Earth imagers: $U_{\text{eff}} = 0.33 \times U_{10\text{m}} + 0.45$. This IME effective wind speed calibration slope which is lower than 1 reflects the fact that the plume extent $L$, defined as the square root of the plume area, is smaller than the actual plume length for long narrow plumes observed over land. This definition of $L$ is chosen for its simplicity and because the plume mask is ventilated by turbulent diffusion rather than uniform transport (Varon et al., 2018). Besides, using this effective wind speed

calibration implicitly assumes that the plume is observed in the same conditions as those used for the LES calibration, including for instance that the full extent of the plume is visible as per the given instrument sensitivity.

### 2.3.2 Effective wind calibration of partial plume observation in IME

The IME method is critically sensitive to the plume mask extent. For a homogeneous plume of $N$ pixels, the source rate $Q$ increases linearly with $\sqrt{N}$. In practice, the plume is not homogeneous and the number of pixels above the instrument

detection threshold relates to the emission rate, and truncating the plume mask because of external factors (low albedo, clouds, etc.) biases $Q$. This IME sensitivity stems from the effective wind speed calibration that relies on an LES sampling of whole plume per the given instrument characteristics. Any systematic plume mask truncation therefore needs to be calibrated for. For the NS2 observation, only the small sea foam patch provides a high enough signal that could allow observation of part of the methane plume, above its source. This specific case therefore requires a custom effective wind calibration.

We consequently re-purpose an ensemble of LES simulations computed for a $275 \times 275$ m$^2$ source area (grossly the NS2 foam patch size) by Maasakkers et al. (2022), scale them to emission rates ranging from 100 to 1000 t/hr, re-sample them according to L8/S-2B instrumental characteristics and perform an effective wind speed calibration that only includes the pixels located above the source area in the plume mask. Following Varon et al. (2021), we perform a linear regression of $U_{\mathrm{eff}}$ against $U_{10\mathrm{m}}$ that is more appropriate for Sentinel-2-like instruments than the logarithm-based regression first proposed in Varon et al. (2018). We obtain the following NS2-custom effective wind speed calibration with an outlier-resilient Huber regression: $U_{\mathrm{eff}} = 1.88 \times U_{10\mathrm{m}} + 0.52$, with a standard deviation of data to fit mismatch values of 1.1 m/s (the Figure supporting this result is provided in the Supplements). This 1.88 calibration factor is significantly different from the slope value given in Sect. 2.3.1, applicable for ideal conditions over land. Its value higher than 1 reflects a different plume definition compared to ideal conditions over land, and must be interpreted as methane excess observed above the area source under-representing the actual emission rate of the full area source. Indeed, only the downwind plume integrates emissions from the all the area source, not the concentration field right above it. Actually, this IME effective wind speed calibration slope close to 2 is consistent with expectations from mass balance of a uniformly ventilated area source (wind direction above it is unique and not changing, a fair assumption at the scale of the NS2 leak) as shown by Buchwitz et al. (2017).

## 2.4 Monte Carlo ensemble approach for evaluating Nord Stream 2 leak rates as seen by Landsat 8 and Sentinel-2B

We use a Monte Carlo ensemble approach to calculate the average methane leak rate from NS2, as seen by L8 and S-2B, using MBSP and IME with our custom calibrations. We consider six different parameters that impact MBSP and/or IME results to generate a Monte Carlo ensemble of leak rate quantifications:

(1) In MBSP, we use the distribution of sea foam albedo spectral dependence calibrations and randomly pick a calibration value from the satellite-wise sets of sea foam observations in ship trails described in Sect. 2.2.2. By doing so, we implicitly follow the underlying distributions of each satellite-wise sea foam spectral dependence calibration values.

(2) To capture the uncertainty in the background, we estimate a non-enhanced methane background over the NS2 sea foam patch. It is computed by applying MBSP using a calibration coefficient exactly equal to the fitted $s_1/s_2$ ratio obtained from the NS2 sea foam pixels, thus compensating for possible methane enhancements. We then compute the standard deviation $\sigma_{X_{CH_4}}$ of this background signal, and use it to randomly shift the MBSP background enhancement by sampling a Gaussian distribution with a standard deviation of $\sigma_{X_{CH_4}}$, and centred on zero.

(3) We vary the plume mask extent by varying the minimum $s_1$ TOA reflectance value for a pixel to be included in the plume mask. These minimum $s_1$ TOA reflectance thresholds sample uniform distributions covering $[0, 0.07]$ for L8 and $[0, 0.045]$ for S-2B. We use different maximum thresholds for each satellite because the maximum TOA reflectance observed by L8 in the NS2 patch is higher than for S-2B (see Fig. 1).

(4) Following Schuit et al. (2023), we include four different 10-m wind speeds to better account for wind speed uncertainty. Three come from meteorological re-analysis products: the European Centre for Medium-Range Weather Forecasts (ECMWF) ERA5 (Hersbach et al., 2020), the Global Forcasting System (GFS) from NOAA National Centers for Environmental Prediction (NCEP, 2000) and the Goddard Earth Observing System-Forward Processing (GESO-FP, Molod et al., 2012). Furthermore,

we include the in-situ wind speed measured at Bornholm airport, which is located about 50 km away from the NS2 leak (IEM, 2023). For September 29[th], we obtain wind speeds of 4.1, 6.6, 4.8 and 3.6 m/s from ERA5, GFS, GEOS-FP and airport measurements, respectively; and for September 30[th], we obtain wind speeds of 5.0, 6.3, 6.3 and 5.7 m/s, respectively. We randomly pick one of these four wind speeds.

(5) To account for wind speed error, we evaluate the differences between the three re-analysis models (ERA5, GEOS-FP, GFS) and in-situ measurements made at Bornholm airport for 2022. On average, we find a standard deviation of 1.6 m/s. We therefore sample the wind speed error from a Gaussian distribution with a 1.6 m/s standard deviation and centred on zero.

(6) We account for effective wind speed calibration errors by randomly sampling data − fit mismatch values from the distribution shown in the Supplements (Figure S8). By doing so, we implicitly follow the slightly non-Gaussian skewed distribution that these mismatches show.

We generate a Monte Carlo ensemble of 1,000,000 members for each satellite overpass, and report their averages, and standard deviations as uncertainty.

Besides these ensemble metrics, we also seek to determine which input parameters contribute most to the obtained ensemble variance. Thus, we also compute the first-order sensitivity indices $S_i$ for our six parameters:

$$S_i = \frac{V_{X_i}\left(E_{\sim X_i}\left(Q|X_i\right)\right)}{V(Q)} \tag{3}$$

with $X_i$ and $i \in \{1,2,3,4,5,6\}$, the six parameters that we explore; $Q$, the leak rates that we compute; $E_{\sim X_i}$ the expectation across all parameters values but $X_i$ that is fixed; $V_{X_i}$, the variance across all $X_i$ values; and $V$, the usual variance. Citing Lo Piano et al. (2021), the plain language meaning of $S_i$ is "the fractional reduction in the variance of $Q$ which would be obtained on average if $X_i$ could be fixed".

Here, we only rely on our single satellite-wise Monte Carlo ensembles and follow Lo Piano et al. (2021) to estimate $S_i$ by directly calculating $V_{X_i}\left(E_{\sim X_i}\left(Q|X_i\right)\right)$ as the variance of the smoothed $Q$ against $X_i$ scatter plot. As we randomly pick values from small sets for the sea foam albedo spectral dependence calibrations and wind speed products, we compute $E_{\sim X_i}\left(Q|X_i\right)$ for each discrete value that $X_i$ can take. For wind speed and effective wind speed errors, as well as minimum albedo and methane enhancement shifts, where we sample continuous distributions, we use 1000 bins of 1000 ensemble members to smooth the Monte Carlo ensemble results.

## 3   Results and discussion: Nord Stream 2 leak rates

Figure 5 shows the distribution of leak rate values within the Monte Carlo ensembles for L8 and S-2B. We obtain ensemble-averaged methane leak rates of $507 \pm 673$ t/hr and $495 \pm 640$ t/hr for L8 and S-2B, respectively. In addition, Table 1 provides the first-order sensitivity indices $S_i$ corresponding to these uncertainties (the smoothed scatter plots supporting these indices are provided in the Supplements). From these indices, we conclude that the uncertainty of the sea foam albedo spectral dependence calibration mainly drives these Monte Carlo ensemble uncertainties. This is illustrated by the colour scale applied to the

255 distributions included in Fig. 5: leak rates get lower and eventually negative with increasing empirical sea foam albedo spectral dependence calibration values.

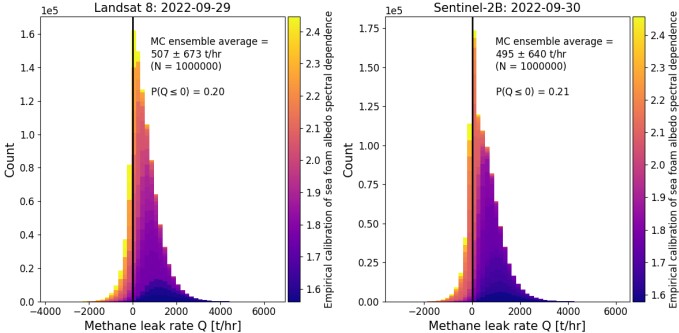

**Figure 5.** Distributions of methane emission rate values for the Landsat 8 (left) and Sentinel-2B (right) ensembles. Monte Carlo ensemble means and standard deviations are shown inset, along with the fraction of null or negative emission rates, denoted as $P(Q \leq 0)$. The color scale shows the contributions of different sea foam albedo spectral dependence calibration values to the overall distribution of leak rates within the ensemble.

| Input parameter | Landsat 8 $S_i$ | Sentinel-2B $S_i$ |
| --- | --- | --- |
| Empirical calibration of sea foam albedo spectral dependence | 0.52 | 0.62 |
| Shift of methane background | 0.18 | 0.12 |
| Minimum albedo to include a pixel in the plume mask | 0.02 | 0.03 |
| Wind speed product | 0.03 | 0.01 |
| Wind speed error | 0.06 | 0.04 |
| Effective wind speed calibration error | 0.01 | 0.01 |

**Table 1.** First-order sensitivity indices $S_i$, computed for each satellite observation and all six parameters included in our Monte Carlo ensembles.

The individual L8 and S-2B ensemble distributions have $\pm 1\sigma$ uncertainty intervals that include zero emissions, and show $P(Q \leq 0) = 0.20$ and $P(Q \leq 0) = 0.21$, respectively. These separate L8 and S-2B estimates may not be independent. For example, similar look-up-tables or IME effective wind calibration errors or biases may hamper them. However, if we oppor-
260 tunistically assume that they are, we can generate an ensemble representing the averaged combined L8 and S-2B NS2 leak rate. We obtain an averaged L8 and S-2B NS2 methane leak rate of $501 \pm 521$ t/hr, with $P(Q \leq 0) = 0.12$. While these uncertainties are too large to draw firm conclusions, we note that both single and dual-overpass estimates show positive means and higher probabilities for positive $Q$ values ($79\% - 88\%$) than negative ones ($12\% - 21\%$). This result hints that L8 and S-2B sensed a methane-related signal, which could be related to an emission magnitude of hundreds of tons per hour.
Because this NS2 observation case is singular and recent, very few results to compare to have been published. GHGSat reports leak rates of 79 t/hr and 29 t/hr for their NS2 glint observations made on Sept 30[th] (GHGSat, 2022). Jia et al. (2022)

report no result for L8, and a methane leak rate of $72 \pm 38$ t/hr for S-2B, while also acknowledging significant uncertainties in their methodology regarding the spectral reflectance of bubbles and the partial imaging of the methane plume. The work performed here precisely describes the origin of the challenges posed by these specific NS2 observations, addresses them through custom calibrations, and provides a comprehensive uncertainty analysis. All previously reported NS2 methane leak rates for September 30[th] are comprised within our large zero-including uncertainty range obtained for S-2B on that day.

## 4  Conclusions

We have evaluated the possibility of extracting methane emission information from Landsat 8 (L8) and Sentinel-2B (S-2B) observations of the Nord Stream 2 (NS2) pipeline leak.

We have shown how the unusual observations of a sea foam patch surrounded by dark still sea (and clouds for L8) challenge implicit underlying assumptions in both the Multi-Band Single-Pass (MBSP) and Integrated Mass Enhancement (IME) methods. For MBSP, we showed that an external empirical calibration of the sea foam albedo spectral dependence is needed, and provided one by using sea foam observations in ship trails. This underlines how extreme surface heterogeneity can hamper the standard albedo spectral dependence calibration in MBSP. For IME, we showed that emission rate quantifications are critically sensitive to plume mask truncation, and we provided an effective wind speed calibration customized to the NS2 leak, for a plume only observed over a small sea foam patch. Plume masks over land can be truncated due to cloud coverage or dark albedo artefacts (waterbodies like rivers, lakes, etc.), which then cause a similar emission rate underestimation.

Using these two-fold customized calibrations for MBSP and IME in a Monte Carlo ensemble approach, we have assessed that no firm conclusion can be made about individual or combined L8 and S-2B detection of the NS2 methane leak. Positive methane leak rates appear to be more likely than negative ones in both single and dual-overpass Monte Carlo ensemble estimates, and point towards a best estimate of $501 \pm 521$ t/hr.

Overall, we see our work as a methodological cautionary tale illustrating how implicit method assumptions need to be considered and compensated for in unusual observation cases such as this one. Our nuanced results with large uncertainties are not surprising: this exceptional Nord Stream leak event pushed Earth imagers that were not initially designed to observe greenhouse gases - even less over water - to their very limits.

*Data availability.* Landsat 8 and Sentinel-2B data used in this work are publicly available and were retrieved from the Google Earth Engine as 2 km-side square images of given targets, from collections `LANDSAT/LC08/C02/T1_TOA` and `COPERNICUS/S2_HARMONIZED`, respectively. All images are listed in the Supplements.

*Author contributions.* MD and JDM conceived the study. MD performed the satellite data analysis and emission rate quantifications, with supervision from JDM and IA. DJV performed the tailored Nord Stream 2 effective wind speed calibration. MD wrote this article with feedback from all co-authors.

*Competing interests.* The authors have the following competing interests: At least one of the (co-)authors is a member of the editorial board of Atmospheric Measurement Techniques.

*Acknowledgements.* This work is in part supported through the ESA funded MethaneCamp project. Copernicus (modified) Sentinel-2 data
(2022) have been used. Authors are grateful to Itziar Irakulis-Loitxate and Otto Hasekamp for the helpful discussions and comments while designing this work. The authors are also grateful to the two referees whose comments helped improve this work.

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
