# Peer review of "Report on Landsat 8 and Sentinel-2B observations of the Nord Stream 2 pipeline methane leak"

_EGUsphere, 2023_

## Referee Comment (RC1)

**Comprehension**

The study examines the feasibility of estimating methane emissions from the Nord Stream 2 (NS2) leak near Bornholm Island in September 2022 using Landsat 8 (L8) and Sentinel-2B (S2B) imager data in two bands of the short wave infrared spectral range. The authors utilize sea foam observations and employed the Multi-Band Single-Pass (MBSP) for the estimation of methane enhancements. For spectral calibration sea foam observations from ship trails are used. For the quantification of leak rates, they use the Integrated Mass Enhancement (IME) method, calibrated for their problem. It is found that even with these adjustments of the MBSP and IME, no firm conclusion can be drawn from individual L8 and S2B detections of the methane leak resulting in large uncertainties in the averaged leak rate estimate.

**General comments:**

In Section 2.3, please include an introductory sentence outlining the methods that could potentially be used for source rate estimation regarding the NS2 problem. After that, explain why IME was selected as the preferred method for quantification.

Please annotate the uncertainties discussed in Section 3 (as well as in the caption of Fig. 5) with the corresponding numbers from Section 2.4.

Consider adding a table that displays the respective 'c' values for the MBSP calibrations. Alternatively, refer to the comments on figures in the specific comments section.

**Specific comments:**

Sec. 2.2:

It's imperative to immediately clarify that the standard approach for the MBSP isn't suitable for the NS2 problem.

In Fig. 1, it should be immediately evident that the variabilities within the L8 and S2B scenes, combined with the CH4-impacted sea foam pixels, are not suitable for MTSB calibration for CH4 detection.

Moreover, given that we anticipate little to no methane signal from dark, still sea pixels, as suggested by Fig. 1, what is the rationale behind including those pixels in the fit for the linear calibration coefficient 'c'?

The term "standard calibration" might be misconstrued. Perhaps consider an alternative term, such as "naïve calibration"?

In my opinion it's crucial to be upfront about the definition and shortcomings of the standard MBSP calibration with respect to this study.

Sec. 2.2.1:

In my assessment, upon reading the section, it immediately becomes evident that the assumption of image-wide pixel calibration, representative for the surface characteristics beneath the plume, is untenable for the context of this study. It should be highlighted right away.

Fig. 1:

Enhance the caption with more detailed information.

It needs clarification that, without adjustments tailored to the NS2-specific challenge ($CH_4$-contaminated sea foam over dark water pixels), the default MBSP calibration falls short of being appropriate.

It's worth noting that no background ($CH_4$-free) sea foam pixels are present in the target scene, as depicted in Fig. 1.

The inclusion of the bottom row of Fig. 1 might be redundant since Fig. 3 already encapsulates that information. Furthermore, the lower panel of Fig. 1 primarily demonstrates an incorrect calibration method for the given context. If it's retained, the caption must be considerably elaborated.

Fig. 1 & 4:

Merging Fig. 1 and Fig. 4 (bottom rows, respectively) into a singular, per-satellite, introductory figure might be a viable approach?

Fig. 3 & 4:

Please ensure, and specify in the caption, that the mean calibrations in Fig. 4 are based on all the ships listed in Tables 2 and 3, respectively.

Furthermore, clarify the rationale behind showcasing ships 1 and 27. Are they particularly unique, or are they simply randomly selected references?

Fig. 5:

Consider to add P(Q) in the caption.

Fig. 2:

Were the dark sea and ship pixels also excluded from the analysis based on the tables provided in the appendix? Please incorporate this detail into the caption.

Fig. 4:

The elevated slope of the NS2 leak patch in Fig. 4, in comparison to the mean calibration from ship foam, seems to be primarily influenced by the notably bright s1 values. This observation becomes more apparent with the distribution of red dots around the fitted red line for lower values; they appear evenly dispersed, and in some instances, seem closer to the blue line. If this observation is accurate, it would be beneficial to note in the caption. This trend could suggest that source attribution might only be feasible for a select number of extremely bright spots (possibly bubbles?), where the Signal-to-Noise Ratio (SNR) is sufficiently high to discern the CH4 enhancement.

l 135-140 (Fig. 3 & 4):

Given the close relation between the bottom plots of Fig. 3 and Fig. 4, it might be prudent to present them within a single figure, divided into four subplots.

l 97:

Following the statement that the calibration strategy implicitly assumes that image-wide pixels are representative of surface characteristics, it's crucial to note that such an assumption is not valid for this particular problem.

l 100:

Please provide some more details on the compilation process of the pre-computed look-up table? Additionally, it would be helpful if you could

provide a reference to the radiative transfer (RT) code or the specific table employed.

l 137:

Perhaps the term "ship foam" should be placed somewhere to remind readers that the average empirical calibration was derived from ship foam observations. Consider to modify the statement to: "... the negative difference of the mean to the ship foam pixel calibration ...".

l 157:

It might be beneficial to mention why Ueff also varies based on the type of observer, especially for Earth-like imagers.

l 207-208:

You choose 10% because the fraction of negative emissions is roughly 10%?

l 203-204 & l 212:

Are you suggesting that the primary source of uncertainty stems from the uncertainties inherent in the imager's observations?

l 212-214:

A sentence for the conclusion?

l 216-217:

It would be beneficial to elaborate further on the statement in parentheses, specifically explaining the reasoning behind the inability to assume independence.

l 218-220:

It would be beneficial to elaborate further on the statement in parentheses, specifically explaining the reasoning behind the inability to assume independence.

l 218:

What does 1M stand for?

Appendix, Table 1:

How is cloud classification defined for S2B? This is crucial, especially considering there will definitely be ship foam pixels where accurate calibration is important.

---

## Referee Comment (RC2)

Review of Dogniaux et al. (submitted to AMT)

This group of authors shows their expertise in the method of estimating emission rates from two imagers, Sentinel-2 and Landsat 8. This is a well-written paper. I have no qualms regarding the use English. I believe the paper should be rejected because of the two main criticisms (immediately below), combined with the fact that the methane emission uncertainties are already very large. If the uncertainties were not so large, I think the paper is sufficiently interesting and meritorious to be published.

Main criticisms:

1) the ensemble approach for estimating the methane emission rate does not cover the correct range of input values. An ensemble should not span $\pm 1\sigma$ because this only covers 68% of the data. I think it would be more appropriate and simpler to only include $+1.0\sigma$ and $-1.0\sigma$ in the ensemble (and not $0\sigma$, or other intermediate increments). Consequently, the authors are underestimating most of their uncertainty sources.
2) The wind speed calibration coefficients should not have an uncertainty of 5%. I also don't see any justification for such a small value. Did it come from 1.88 versus 2 on L172, which is a 6% difference?

Fig. 1: Please add the dates to the caption for the two reflectance images. Latitude and longitude tick marks would be of interest to the readers.

L107: The linear calibration coefficient varies strongly between Landsat and Sentinel-2. I wonder if these studies are even relevant? Was the reflectance from the bubble monolayer or the multiple-bubble layer used in the Whitlock et al. study? Do they give the same ratio? I could not access the Koepke paper (but the reference is correct). The authors should note that this reflectance ratio should be roughly the same from space and at the ground because the atmosphere is optically thin. On second thought, is (background) methane a strong enough absorber to affect the ratio (satellite versus ground-based)? Whitlock et al. used a ground-based radiometer.

L132: "through" seems incorrect and terse. I suggest "via the use of"

L151: "mask" could be deleted for simplicity

L172: This linear regression equation has a much different slope than the one in Fig. 4 of Varon et al. (2018). Was $\log(U_{10})$ also tried? There should be more discussion of why the effective wind speed should exceed $U_{10}$ for this foamy setting. I think a 5% uncertainty (L197) on the effective wind speed relationship is a gross underestimate. Given the effective wind speed equation on L157, the authors should greatly expand the magnitude of this sixth source of error/uncertainty, maybe by an order of magnitude.

L173: What is the relevance of the uniformity of the ventilation?

L189: Why include four speeds and perturb each of them by 50%? The authors might actually be overestimating this source of uncertainty, since L193-L195 show that the wind speed range is not that large, especially on Sept. 30th.

L198: I cannot reproduce the numbers.

L218: Re: "1M", is this 1 million? If so, please avoid the shorthand and I don't see why the random draws did not come from ~5 million members, but it might not matters. I simply need to know how the one million members were selected by the authors in order to assess whether this is a biased sample.

L236 (and in the abstract): hypotheses-> assumptions

---

## Author Comment (AC1)

We are grateful to the referee for the very detailed feedback and interest in our work. Our answers to the comments and questions are provide below in red.

Comprehension

The study examines the feasibility of estimating methane emissions from the Nord Stream 2 (NS2) leak near Bornholm Island in September 2022 using Landsat 8 (L8) and Sentinel-2B (S2B) imager data in two bands of the short wave infrared spectral range. The authors utilize sea foam observations and employed the Multi-Band Single-Pass (MBSP) for the estimation of methane enhancements. For spectral calibration sea foam observations from ship trails are used. For the quantification of leak rates, they use the Integrated Mass Enhancement (IME) method, calibrated for their problem. It is found that even with these adjustments of the MBSP and IME, no firm conclusion can be drawn from individual L8 and S2B detections of the methane leak resulting in large uncertainties in the averaged leak rate estimate.

General comments:

In Section 2.3, please include an introductory sentence outlining the methods that could potentially be used for source rate estimation regarding the NS2 problem. After that, explain why IME was selected as the preferred method for quantification.

We added this introductory sentence and justified our choice of the IME.

| New text: line 160 – 164 | 2.3.1 General description

If a plume is observed in an image resulting from MBSP, the associated emission rate can be quantified using **different approaches such as the Gaussian plume inversion (GP), source pixel (SP), Cross-Section flux (CSF) and Integrated Mass En- hancement (IME) methods (Varon et al., 2018). Because GP and SP are not suited for the quantification of plumes detected using high-resolution satellite observations, and the CSF relies on several transects drawn on an extended downwind plume, we use the IME method.** |
|---|---|

Please annotate the uncertainties discussed in Section 3 (as well as in the caption of Fig. 5) with the corresponding numbers from Section 2.4.

We understand this comment as asking to relate uncertainties obtained for methane leak rates to the width of the interval explored by the ensemble for each of the six input parameters.

Following Referee 2's comments, we modified this ensemble quantification to a full Monte Carlo ensemble quantification, thus considering the actual distributions of the input parameters that we explore to assess methane leak rate uncertainties. In addition to this new way of generating the ensemble, we also compute the first-order sensitivity indices for all six input parameters. They describe the contribution of each input parameter variance to the methane leak rate variance. The obtained first-order sensitivity indices answer this question of relating input parameter uncertainty to methane leak rate uncertainty.

The method that we employ to compute the indices is now detailed in an expanded and revised Sect 2.4, and the indices and conclusions that follow are given at the beginning of Sect. 3 in the revised manuscript. The Supplements now also include intermediate result plots illustrating the calculation of these indices.

Consider adding a table that displays the respective 'c' values for the MBSP calibrations. Alternatively, refer to the comments on figures in the specific comments section.

We understand that this general comment is related to the specific comment on Figures 3 & 4: "Please ensure, and specify in the caption, that the mean calibrations in Fig. 4 are based on all the ships listed in Tables 2 and 3, respectively", and that the table discussed here would include all 27 and 38 individual satellite-wise c values for L8 and S-2B, respectively.

We have expanded the Tables in the Supplements to include a column that provides the c calibration coefficient for each ship observation (which were already printed in the scatter plots included in the Supplements), and we have modified the text in Sect 2.2.2 and in captions of Fig 3 and 4 to refer the reader to the Supplements.

| New text: line 135 | [...] to determine $c_i$, the coefficient describing the spectral dependence of sea foam albedo for the i-th image (see individual **$c_i$ values and fits obtained for each ship trail observation** in the Supplements). |
|---|---|
| New text: Fig. 3 | Empirically determined sea foam albedo spectral dependence between s1 and s2 for Landsat 8 (left) and Sentinel-2B (right). Sea foam pixels for all ship images are depicted (**dots with different colors indicating different ships**, the legend only includes elements for the first and last images), along with their respective calibration slopes (**thin lines, each is detailed in the Supplements,** the legend only includes elements for the first and last images)**.** |
| New text: Fig. 4 | Comparisons of s1 and s2 TOA reflectance (bottom) depicting different pixel types and showing the empirically determined spectral dependence of sea foam albedo (thick blue line, **the individual ship trail observations underlying this result are shown in Fig. 3 and in supplementary Tables S2 and S3 for L8 and S-2B, respectively)**, and the s1/s2 ratio observed over the NS2 sea foam patch (red line). |

Specific comments:

Sec. 2.2:

It's imperative to immediately clarify that the standard approach for the MBSP isn't suitable for the NS2 problem.

We now included a short introduction to Sect 2.2 that already announces that the usual MBSP calibration will prove to be unsuitable for this specific NS2 case study. Symmetrically, we have also written a short introduction to Sect. 2.3.

| New text: line 92 – 94 | 2.2 Methane enhancement retrieval: the Multi-Band Single-Pass (MBSP) method

**We use the Multi-Band Single-Pass (MBSP) method to retrieve local methane column enhancements from Earth imager observations. We first describe MBSP and its standard calibration approach, and then show how this specific NS2 case study calls for a custom calibration.** |

| | 2.2.1 General description |
|---|---|
| New text: line 156 – 158 | 2.3 Emission rate quantification: the Integrated Mass Enhancement (IME) method

**We use the Integrated Mass Enhancement (IME) method to quantify the methane emission rate from local methane column enhancement retrievals that show an emission plume. Here, we first explain why we choose the IME method and how it works, then we explain why this specific NS2 case study also calls for a custom calibration for the IME method.**

2.3.1 General description |

In Fig. 1, it should be immediately evident that the variabilities within the L8 and S2B scenes, combined with the CH4-impacted sea foam pixels, are not suitable for MTSB calibration for CH4 detection.

The new short introduction to Sect. 2.2 now provides this information (see above). Besides, we have added this information in the caption of Fig. 1 as well, as recommended later in this review.

Moreover, given that we anticipate little to no methane signal from dark, still sea pixels, as suggested by Fig. 1, what is the rationale behind including those pixels in the fit for the linear calibration coefficient 'c'?

At the stage of Figure 1 in the paper, the rationale is to illustrate the "standard" naïve MBSP calibration strategy by actually applying it as it was presented in Varon et al. (2021) on the data at hand for L8 and S-2B, meaning on the full image (as it was presented), and not just the sea foam pixels. We have changed the text to better reflect this rationale.

| New text: line 123 – 125 | This issue similarly applies to the L8 NS2 observation, that also features an additional complication: very bright clouds are present in the image, which in this case drive the standard MBSP calibration (c = 1.13). **Thus, the standard MBSP calibration lines included in Fig. 1 illustrate why t**he NS2 observation case, that relies on a small sea foam patch, calls for an external calibration of the spectral dependence of sea foam albedo. |
|---|---|

The term "standard calibration" might be misconstrued. Perhaps consider an alternative term, such as "naïve calibration"?

We agree that the standard calibration employs a naïve approach, and have included this adjective in a few places in the revised manuscript to describe it in Sect 2.2.1. In addition, we also specified that it will be "hereafter referred to as 'standard MBSP calibration'".

| New text: line 102 – 105 | **This calibration strategy was proposed with the MBSP method by Varon et al. (2021),** and implicitly assumes that image-wide pixels are representative of the surface characteristics expected below the (potential) methane plume. **Hereafter, we will refer to this "naïve" calibration strategy as the "standard MBSP calibration".** The rationale of MBSP is that deviations in the methane-sensitive s2 band [...] |
|---|---|

In my opinion it's crucial to be upfront about the definition and shortcomings of the standard MBSP calibration with respect to this study.

The short introduction that we added to Sect. 2.2 addresses this question of being upfront about the shortcomings that are going to be described in the subsection (see above).

Sec. 2.2.1:
In my assessment, upon reading the section, it immediately becomes evident that the assumption of image-wide pixel calibration, representative for the surface characteristics beneath the plume, is untenable for the context of this study. It should be highlighted right away.
The short introduction that we added to Sect. 2.2 addresses this question of highlighting right away that the standard MBSP calibration will prove unsuitable for this specific NS2 case, with explanations given later in Sect. 2.2.2 (see above).

Fig. 1:

Enhance the caption with more detailed information.
We revised the caption to provide more detailed information.

| New text: Fig. 1 | Landsat 8 (left, **September 29th 2022**) and Sentinel-2B (right, **September 30th 2022**) images of the Nord Stream 2 leak for s1 (top), and s1 and s2 TOA reflectance comparisons depicting different pixel natures and showing the standard MBSP c calibration line (bottom). **The pixel natures of dark still sea (black), clouds (grey) and NS foam patch (red, all influenced by the methane leak) are separated using empirically determined thresholds given in the Supplements. The standard MBSP calibration (dashed line) is provided here to illustrate why it proves to be unsuitable for this specific NS2 case, as detailed in Sect. 2.2.2.** |
|---|---|

It needs clarification that, without adjustments tailored to the NS2-specific challenge (CH4-contaminated sea foam over dark water pixels), the default MBSP calibration falls short of being appropriate.
We have elaborated the Fig. 1 caption following this comment. It now explains that this standard calibration will prove unsuitable for this case and refers to the discussion of this point in Sect. 2.2.2 (see above).

It's worth noting that no background (CH4-free) sea foam pixels are present in the target scene, as depicted in Fig. 1.
This point is indeed discussed in Sect. 2.2.2 when describing why the standard calibration is unsuitable for this case study. We added an element to the caption of Fig. 1 to reflect this aspect: "NS foam patch (red, all influenced by the methane leak)", see above.

The inclusion of the bottom row of Fig. 1 might be redundant since Fig. 3 already encapsulates that information.
Information is slightly redundant indeed, but Figures 1, 3 and 4 have been designed to follow a progression to better underline the different steps of the work we perform.
- Figure 1 provides a first candid look at the data, thus includes the naïve standard calibration which is now commented upon in the caption and helps to clarify why this standard approach is not suitable for this case.

- Figure 3 is dedicated to the empirical calibration using ship trail observations. It still contains the standard calibration to show the reader how they compare to the ship-based calibration.
- Figure 4 is dedicated to the methane enhancement retrieval part of MBSP, thus contains the NS2 pixels points and averaged empirical ship-based calibrations, which is the one we use to calibrate the MBSP for the methane enhancement retrieval.

We feel that merging or breaking these figures apart would confuse the progression that happens between these three figures.

Furthermore, the lower panel of Fig. 1 primarily demonstrates an incorrect calibration method for the given context. If it's retained, the caption must be considerably elaborated.
We have elaborated the Fig. 1 caption following this comment. It now explains that this standard calibration will prove unsuitable for this case and refers to the discussion of this point in Sect. 2.2.2 (see above).

Fig. 1 & 4:

Merging Fig. 1 and Fig. 4 (bottom rows, respectively) into a singular, per-satellite, introductory figure might be a viable approach?
Please refer to the answer above regarding the progression between Figures 1, 3 and 4.

Fig. 3 & 4:

Please ensure, and specify in the caption, that the mean calibrations in Fig. 4 are based on all the ships listed in Tables S2 and S3, respectively.
We have adjusted the caption of Fig. 4 in revised manuscript regarding this comment.

| New text: Fig. 4 | Comparisons of s1 and s2 TOA reflectance (bottom) depicting different pixel types and showing the empirically determined spectral dependence of sea foam albedo (thick blue line, **the individual ship trail observations underlying this result are shown in Fig. 3 and in supplementary Tables S2 and S3 for L8 and S-2B, respectively)**, and the s1/s2 ratio observed over the NS2 sea foam patch (red line). |
| --- | --- |

Furthermore, clarify the rationale behind showcasing ships 1 and 27. Are they particularly unique, or are they simply randomly selected references?
Ships 1 and 27 are shown in Fig. 3 for L8 as are shown ships 1 and 38 for S-2B, because they are the first and last ship trail observations included in the data sets for L8 and S-2B, sorted in chronological order. All sea foam observations pixels and fits are shown in Fig. 3, but the legend itself only includes the first and last observations that bound the sets and pink-to-yellow colormaps. The three dots "…" after 'Fitted ratio (Ship 1)' are included to represent this idea. The caption of Fig 3 has been adjusted in the revised manuscript to better explain this.

| New text: Fig. 3 | Empirically determined sea foam albedo spectral dependence between s1 and s2 for Landsat 8 (left) and Sentinel-2B (right). Sea foam pixels for all ship images are depicted (**dots with different colors indicating different ships, the legend only includes elements for the first and last images)**, along with their respective calibration slopes (**thin lines, each is detailed in the Supplements, the legend only includes elements for the first and last images).** |
| --- | --- |

Fig. 5:

Consider to add P(Q) in the caption.
We added the P(Q ≤ 0) notation explanation in the Fig. 5 caption.

| New text: Fig. 5 | Distributions of methane emission rate values for the Landsat 8 (left) and Sentinel-2B (right) ensembles. **Monte Carlo** ensemble means and standard deviations are shown inset, along with the fraction of null or negative emission rates, **denoted as P (Q ≤ 0).** The color scale shows the contributions of different sea foam albedo spectral dependence calibration values to the overall distribution of leak rates within the ensemble. |
|---|---|

Fig. 2:

Were the dark sea and ship pixels also excluded from the analysis based on the tables provided in the appendix? Please incorporate this detail into the caption.
Yes, they were, as detailed in the text (line 123 in the original manuscript). We have added this explanation in the Fig. 2 caption as well in the revised manuscript.

| New text: Fig. 2 | Example of sea foam observation in the Sentinel-2B image of a ship trail acquired on October 12th, 2022. Dark still sea and ship pixels have been removed and are shown in grey and white, respectively. **They are also excluded from the sea foam albedo spectral dependence results presented later in Sect. 2.2.2 and in the Supplements.** |
|---|---|

Fig. 4:

The elevated slope of the NS2 leak patch in Fig. 4, in comparison to the mean calibration from ship foam, seems to be primarily influenced by the notably bright s1 values. This observation becomes more apparent with the distribution of red dots around the fitted red line for lower values; they appear evenly dispersed, and in some instances, seem closer to the blue line. If this observation is accurate, it would be beneficial to note in the caption. This trend could suggest that source attribution might only be feasible for a select number of extremely bright spots (possibly bubbles?), where the Signal-to-Noise Ratio (SNR) is sufficiently high to discern the CH4 enhancement.

Indeed, the difference between the ship trail-based calibration and the s1/s2 fitted line on NS2 sea foam pixels is less apparent for the lowest sea foam albedo pixels in the NS2 images. When we get closer to the center of foam patch, it becomes brighter, thus potentially enabling to better distinguish the absorbing impact of methane on the s2 band, which leads to lower s2 values than the empirical calibration (blue) line, that thus results in pixels being above (and left) of the ship-based calibration (blue) line. So, we agree with this "higher SNR" interpretation of the "brighter foam patch center", and have extended the Fig. 4 caption to include this idea.

| New text: Fig. 4 | Comparisons of s1 and s2 TOA reflectance (bottom) depicting different pixel types and showing the empirically determined spectral dependence of sea foam albedo (thick blue line, **the individual ship trail observations underlying this result are shown in Fig. 3 and in supplementary Tables S2 and S3 for L8 and S-2B, respectively)**, and the s1/s2 ratio observed over the NS2 sea foam patch (red line). **The higher slopes shown by the s1/s2 ratios (red) compared to the empirical calibrations (blue) are driven by the brightest pixels at the center of the sea foam patch that offer a better signal-to-noise ratio to observe methane absorption than darker pixels.** |
|---|---|

l 135-140 (Fig. 3 & 4):

Given the close relation between the bottom plots of Fig. 3 and Fig. 4, it might be prudent to present them within a single figure, divided into four subplots.

Please refer to the answer above regarding the progression between Figures 1, 3 and 4.

l 97:

Following the statement that the calibration strategy implicitly assumes that image-wide pixels are representative of surface characteristics, it's crucial to note that such an assumption is not valid for this particular problem.

The purpose of Sect 2.2.1 is to describe MBSP in the general case, as it was first presented in Varon et al. (2021). This comment has been addressed by including a short introduction to Sect. 2.2 which announces that the standard calibration of MBSP will prove to be unsuitable for this NS2 case (see above).

l 100:

Please provide some more details on the compilation process of the pre- computed look-up table? Additionally, it would be helpful if you could provide a reference to the radiative transfer (RT) code or the specific table employed.

We have added an extra sentence describing the input atmosphere and spectroscopic database (HITRAN 2020) that we employ to generate the look-up-tables.

| New text: line 108 – 110 | The translation of ΔR to methane enhancements is performed using pre-computed look-up tables, generated through radiative transfer simulations. **Here, they are based on the 2020 version of the HITRAN spectroscopic database (Gordon et al., 2022), rely on a 21-layer atmospheric model representative of mid-latitudes and include the impact of the solar zenith angle.** |
|---|---|

l 137:

Perhaps the term "ship foam" should be placed somewhere to remind readers that the average empirical calibration was derived from ship foam observations. Consider to modify the statement to: "... the negative difference of the mean to the ship foam pixel calibration ...".

We adjusted the sentenced in the revised manuscript as suggested.

| New text: line 148 | This **ship-based $\bar{c}$ –s1/s2** negative difference overall translates to positive methane enhancement through MBSP. |
|---|---|

l 157:

It might be beneficial to mention why Ueff also varies based on the type of observer, especially for Earth-like imagers.
This is already implicitly mentioned in line 156 of the original manuscript, when explaining that LES simulations have to be resampled according to instrument characteristics (spatial resolution, noise, etc). We have added an extra sentence to better reflect this comment.

| New text: line 174 – 175 | Plume transport includes complicated three-dimensional and turbulent effects that require computer-intensive simulations to be accounted for, if even possible given the randomness of turbulence. Through IME, the overall impacts of those effects are presumably captured into a single effective wind speed, denoted Ueff. Ueff is calibrated against the 10-m wind speed provided by meteorological models (U10m) over a set of Large Eddy Simulations (LES) made for known synthetic emission rates, and re-sampled according to a given instrument characteristics (spatial resolution, noise model, etc.). **Thus, Ueff can be calibrated for specific instruments and observing conditions.** Varon et al. (2021) provide an effective wind speed calibration model for Sentinel-2-like Earth imagers: Ueff = 0.33 × U10m + 0.45. |
|---|---|

l 207-208:

You choose 10% because the fraction of negative emissions is roughly 10%?
10% was rather an arbitrary symbolic threshold with no justification. We do not employ it anymore in the revised manuscript (see revised manuscript Sect. 3).

l 203-204 & l 212:

Are you suggesting that the primary source of uncertainty stems from the uncertainties inherent in the imager's observations?
The primary source of uncertainty is the uncertainty on the spectral dependence of sea foam albedo. In the revised manuscript, it is now clearly shown thanks to the calculation of first order sensitivity indices. In line 212 in the original manuscript, we report "uncertainties" (methodological drawback may be a more appropriate expression) as written in Jia et al. (2022).

l 212-214:

A sentence for the conclusion?
A similar message in developed in a longer piece of text in the conclusion indeed. We think this sentence is relevant here as part of the discussion to explain that we explored the methodological drawbacks acknowledged by Jia et al (2022).

l 216-217:

It would be beneficial to elaborate further on the statement in parentheses, specifically explaining the reasoning behind the inability to assume independence.

We adjusted the revised manuscript to develop the reason why the quantifications may not be independent in a sentence before this one. The reason is that both satellite observations are processed with Look-up-tables that can for example be hampered by similar spectroscopy error originating from the HITRAN 2020 database itself, or by errors coming from the fact that IME estimates rely on the same set of LES simulations.

| New text: line 258 – 259 | […] have ±1σ uncertainty intervals that include zero emissions, and show P (Q ≤ 0) = 0.20 and P (Q ≤ 0) = 0.21, respectively. **These separate L8 and S-2B estimates may not be independent. For example, similar look-up-tables or IME effective wind calibration errors or biases may hamper them.** However, if we opportunistically assume that they are, we can generate an ensemble […] |
|---|---|

l 218-220:

It would be beneficial to elaborate further on the statement in parentheses, specifically explaining the reasoning behind the inability to assume independence.
Please refer to the answer to the previous item.

l 218:

What does 1M stand for?
It stands for 1 million, we stopped using this notation in the revised manuscript.

Appendix, Table 1:

How is cloud classification defined for S2B? This is crucial, especially considering there will definitely be ship foam pixels where accurate calibration is important.
There is no "cloud classification" performed for S-2B as there are no clouds to remove from this image of the NS2 leak acquired by S-2B on Sept. 30th 2022. We adjusted the table to state that no cloud filtering is needed for S-2B.

| New text: Table S1 | **No cloud filtering required for this S-2B image** |
|---|---|

All sea foam images in ship trails have been chosen so that cloudy pixels are not included in the s1 against s2 fits. This can be easily verified by examining Figures included in the Supplements: the vast majority of the pixels (dots) satisfactorily align with a s1/s2 = 1.8-2.0 slope, which is far from the cloud-related s1/s2 = 1.13 slope shown by the standard L8 calibration that was driven by cloudy pixels before the empirical ship-based calibration (see Fig. 1).

References

Varon, D. J., Jervis, D., McKeever, J., Spence, I., Gains, D., and Jacob, D. J.: High-frequency monitoring of anomalous methane point sources with multispectral Sentinel-2 satellite

observations, Atmospheric Measurement Techniques, 14, 2771–2785, https://doi.org/10.5194/amt-14-2771-2021, 2021

Jia, M., Li, F., Zhang, Y., Wu, M., Li, Y., Feng, S., Wang, H., Chen, H., Ju, W., Lin, J., Cai, J., Zhang, Y., and Jiang, F.: The Nord Stream pipeline gas leaks released approximately 220,000 tonnes of methane into the atmosphere, Environmental Science and Ecotechnology, 12, 100 210, https://doi.org/https://doi.org/10.1016/j.ese.2022.100210, 2022.

---

## Author Comment (AC2)

We are grateful to the referee for this stimulating feedback that brought important additions to our work. Our answers to the comments and questions are provided in red below.

Review of Dogniaux et al. (submitted to AMT)

This group of authors shows their expertise in the method of estimating emission rates from two imagers, Sentinel-2 and Landsat 8. This is a well-written paper. I have no qualms regarding the use English. I believe the paper should be rejected because of the two main criticisms (immediately below), combined with the fact that the methane emission uncertainties are already very large. If the uncertainties were not so large, I think the paper is sufficiently interesting and meritorious to be published.

We agree that uncertainties are large indeed. However, they are actually part of our motivation to submit this work to AMT.

Our work is related to rather novel Earth-imager data exploitation techniques that are easy to implement, already brought very relevant scientific results, and are currently gaining significant momentum within the greenhouse gas remote sensing community. When the Nord Stream leak happened, these techniques or closely-related ones were very swiftly applied to process L8 and S-2B observations which resulted in very rapid communications (on the same day by the International Methane Emissions Observatory: https://twitter.com/MethaneData/status/1575610350548164608) and a Short Communication submitted to "Environmental Science and Ecotechnology" two weeks later (Jia et al, 2022). These communications and other (for example a poster at the AGU 2022 Fall Meeting) claimed that a methane plume had been detected by Landsat 8 and Sentinel-2B, and Jia et al (2022) even provided an emission estimate for the Sentinel-2B observation of 72 ± 38 t/hr, while also acknowledging significant and not so well-defined "uncertainties" ("methodological drawback" may be a more appropriate expression) related to the reflectance of bubbles and part of the plume missing from the observation.

In this submission to AMT, we exactly explain what aspects of the novel techniques are challenged by the Nord Stream 2 observation case, and include a comprehensive uncertainty analysis (improved following this review) that strongly nuances what can be stated about the methane leak based on L8 and S-2B. Therefore, our study can serve as a methodological cautionary tale and detailed discussion of these novel techniques. This is especially important as results from the methane community are increasingly used by a growing group of stakeholders including national governments and international NGOs. We therefore partly revised our Abstract, Sect. 3 and Conclusions to better underline the significance of our work as providing important insight on often used methods.

| New text: line 20 – 26 | **Our comprehensive uncertainty analysis yields large methane leak rate uncertainty ranges that include zero, with a best estimate of 501±521 t/hr. Thus, no firm conclusion can be drawn from the single or combined overpasses of L8 and S-2B. Within all our Monte Carlo ensembles, positive methane leak rates have higher probabilities (79 – 88%) than negative ones (12 – 21%), thus indicating that L8 and S-2B likely captured a methane-related signal. Overall, we see our work both as a nuanced analysis of L8 and S-2B contributions to quantifying the NS2 leak emissions** |

| | **and as a methodological cautionary tale that builds insight on MBSP and IME sensitivities.** |
|---|---|

Main criticisms:
1) the ensemble approach for estimating the methane emission rate does not cover the correct range of input values. An ensemble should not span ±1sigma because this only covers 68% of the data. I think it would be more appropriate and simpler to only include +1.0 sigma and -1.0 sigma in the ensemble (and not 0 sigma, or other intermediate increments). Consequently, the authors are underestimating most of their uncertainty sources.

We recognize that the "grid-based" approach we used to build the ensemble falls short in grasping the impact of the full variability of our input parameters. Consequently, we revised our approach and set up a Monte Carlo ensemble that follows best-estimate distributions for all of our six input parameters. The revised uncertainty estimation approach is described in Sect. 2.4 of the revised manuscript. Resulting uncertainties are now ~40% higher in standard deviation, thus confirming that the uncertainties were underestimated However, as explained above, we do think that the size of these uncertainties underscore the scientific significance of our work. The updated uncertainty values are included in section 3.

2) The wind speed calibration coefficients should not have an uncertainty of 5%. I also don't see any justification for such a small value. Did it come from 1.88 versus 2 on L172, which is a 6% difference?

Within the scope of our Monte Carlo ensemble approach, we now use the actual distribution of fit – data mismatches (standard deviation of 1.1 m/s) to perturbate the calculated effective wind speed. This 1.1 m/s standard deviations amounts to 12% and 10% of the average effective wind speed that we compute for L8 and S-2B overpasses. This distribution is given in an additional figure included in the Supplements (Fig. 13).

| New text: line 231 – 233 | **(6) We account for effective wind speed calibration errors by randomly sampling data – fit mismatch values from the distribution shown in the Supplements (Figure S8). By doing so, we implicitly follow the slightly non-Gaussian skewed distribution that these mismatches show.** |
|---|---|

Fig. 1: Please add the dates to the caption for the two reflectance images. Latitude and longitude tick marks would be of interest to the readers.
We added central latitude and longitude ticks, and dates in all satellite image panels included in the paper. The captions have been updated as well and include dates in the revised manuscript.

L107: The linear calibration coefficient varies strongly between Landsat and Sentinel-2. I wonder if these studies are even relevant? Was the reflectance from the bubble monolayer or the multiple-bubble layer used in the Whitlock et al. study? Do they give the same ratio? I could not access the Koepke paper (but the reference is correct). The authors should note that this reflectance ratio should be roughly the same from space and at the ground because

the atmosphere is optically thin. On second thought, is (background) methane a strong enough absorber to affect the ratio (satellite versus ground-based)?
Whitlock et al. used a ground-based radiometer.

These studies were the only reference that we could find to benchmark our space-based results using L8 and S-2B satellite observations. The PDFs that we could find are digitized copies of on-paper versions, and plot quality at the time did not help reading exactly their results. We could evaluate through graphical reading (as written in the original version) that between 1.6 μm and 2.2 μm, the ratio in spectral reflectance should be about 2 or a little lower. Please find below screenshots of the graphs we used.

**Whitlock et al, 1982 (we read the multiple layers curve)**
**https://agupubs.onlinelibrary.wiley.com/doi/abs/10.1029/GL009i006p00719**

[Figure]

(a) Reflectance of foam patches.

**Keopke et al, 1984**
**https://www.researchgate.net/publication/243574607_Effective_reflectance_of_oceanic_whitecps**

[Figure]

Fig. 8.    Effective reflectance of oceanic foam as a function of wave-
length, after Eq. (9).

The spectral dependence of reflectance that we assess between these satellite bands are rather the spectral dependence of Top-of-the-atmosphere reflectance, or 'space-effective', because the atmosphere is not optically thin enough to transmit light seamlessly to its top. As shown in Varon et al (2021), these 1.6 and 2.2 µm bands also exhibit spectral lines of water vapor for instance. On the top of these lines, continuum absorption of water vapor, which varies by about one order of magnitude between these two bands, must be added (Shine et al, 2016), and we have seen for other imagers impacts of image-wide water vapor gradients on band ratios that are used to retrieve methane enhancements. The scatter in fitted s1/s2 ratio between ship observations for similar satellites might be explained by water vapor variability for instance. As hypothesized in the comment, methane background concentration variability might also play a role. Other causes could be bi-directional reflectance effects due to slightly different viewing and sun angle geometries (we focused our search around the North and Baltic Seas and in the month preceding and following the NS2 leak to minimize these effects).

Differences between the calibrations obtained for L8 and S-2B could be explained by the small – but existing – differences in the definition intervals of the bands they observe, as well as by their narrow band filter shapes (Zhang et al, 2018). Finally, fewer cases could be observed for L8 because its 30x30 m² resolution does not allow to observe as many satisfying ship trails as for S-2B which has a 20x20 m² resolution.

All these aspects may explain the variability of s1/s2 ratios obtained for each satellite, and the slight difference in satellite-averaged results. Their precise discussion is however outside the scope of this study.

We added additional explanations on the TOA and ground-based difference between our work and the references in the revised manuscript.

| New text:
line 138 –
139 | […] We obtain $\bar{c}$ = 1.96 ± 0.23 and $\bar{c}$ = 1.91 ± 0.22 for L8 and S-2B, respectively. **These Top-of-the-Atmosphere reflectance ratios are overall** consistent with results presented by Whitlock et al. (1982) and Koepke (1984) **that were measured on the ground.** Comparing the S-2B result to the slightly higher standard MBSP […] |
| --- | --- |

L132: "through" seems incorrect and terse. I suggest "via the use of"
We corrected this text as suggested.

| New text:
line 143 | […] reduces ΔR = (cs2 − s1 )/s1 , which translates to an increase of methane enhancement **via the use of MBSP**. |
| --- | --- |

L151: "mask" could be deleted for simplicity
We delete "mask" in the revised manuscript.

| New text:
line 169 | […] the plume length, XCH4i, the total column methane enhancement of the i-th **plume pixel**, and ai, the area of this pixel. |
| --- | --- |

L172: This linear regression equation has a much different slope than the one in Fig. 4 of Varon et al. (2018). Was log(U10) also tried? There should be more discussion of why the effective wind speed should exceed U10 for this foamy setting. I think a 5% uncertainty (L197) on the effective wind speed relationship is a gross underestimate. Given the effective wind speed equation on L157, the authors should greatly expand the magnitude of this sixth source of error/uncertainty, maybe by an order of magnitude.

We followed Varon et al. (2021) who prescribe a linear effective wind calibration for Sentinel-2 specifically, and thus did not try the log(U10) calibration.

| New text:
line 193 –
195 | […] and perform an effective wind speed calibration that only includes the pixels located above the source area in the plume mask. **Following Varon et al. (2021), we perform a linear regression of Ueff against U10m that is more appropriate for Sentinel-2-like instruments than the logarithm-based regression first proposed in Varon et al. (2018).** We obtain the following NS2-custom effective wind speed calibration with an outlier-resilient […] |
| --- | --- |

In the revised manuscript, we expanded on the discussion of why the effective wind speed should exceed U10 in this specific case. These additions explain why in ideal conditions over land the effective wind speed slope is below 1, and why it is above 1 for the NS2 case

| New text:
line 176 –
179 | Varon et al. (2021) provide an effective wind speed calibration model for Sentinel-2-like Earth imagers: Ueff = 0.33 × U10m + 0.45. **This IME effective wind speed calibration slope which is lower than 1 reflects the fact that the plume extent L, defined as the square root of the plume area, is smaller than the actual plume length for long narrow plumes observed over land. This definition of L is chosen for its simplicity and because the plume mask is ventilated by turbulent diffusion rather than uniform transport (Varon et al., 2018).** Besides, using this effective wind speed calibration implicitly assumes that the plume is observed in the same conditions as those used for the LES calibration, including for instance that the full extent of the plume is visible as per the given instrument sensitivity. |
| --- | --- |

| New text: line 197 – 201 | This 1.88 calibration **factor is significantly different from the slope value given in Sect. 2.3.1, applicable for ideal conditions over land. Its value higher than 1 reflects a different plume definition compared to ideal conditions over land, and must be interpreted as methane excess observed above the area source under-representing the actual emission rate of the full area source. Indeed, only the downwind plume integrates emissions from the all the area source, not the concentration field right above it.** Actually, this IME effective wind speed calibration slope close to 2 is consistent with expectations from mass balance of a uniformly ventilated area source (wind direction above it is unique and not changing, a fair assumption at NS2 leak scale) as shown by Buchwitz et al. (2017). |
|---|---|

Besides, we now provide in the Supplements the scatter plot of LES sampling points and Huber linear fit that were performed to obtain this effective wind speed calibration (Figure 13). It also gives the distribution of effective wind speed – fit mismatches which shows a 1.1 m/s standard deviation. This represents 12% and 10% of the averaged effective wind speed we obtain from the values sampled from ERA5, GEOS-FP, GFS and that we get from Bornholm airport for L8 and S-2B overpasses, respectively.

| New text: line 231 – 233 | **(6) We account for effective wind speed calibration errors by randomly sampling data – fit mismatch values from the distribution shown in the Supplements (Figure S8). By doing so, we implicitly follow the slightly non-Gaussian skewed distribution that these mismatches show.** |
|---|---|

L173: What is the relevance of the uniformity of the ventilation?
We clarified this point in the revised manuscript.

| New text: line 202 – 203 | Actually, this IME effective wind speed calibration slope close to 2 is consistent with expectations from mass balance of a uniformly ventilated area source (**wind direction above it is unique and not changing, a fair assumption at the scale of the NS2 leak)** as shown by (Buchwitz et al., 2017). |
|---|---|

L189: Why include four speeds and perturb each of them by 50%? The authors might actually be overestimating this source of uncertainty, since L193-L195 show that the wind speed range is not that large, especially on Sept. 30th.

We include four different wind speeds because this gives a better representation of the actual uncertainty in the wind compared to just perturbating one value. We adjusted the revised manuscript.

| New text: line 220 | (4) **Following Schuit et al. (2023),** we include four different 10-m wind speeds **to better account for wind speed uncertainty.** Three come from meteorological re-analysis products: the European Centre for Medium-Range Weather Forecasts (ECMWF) ERA5 (Hersbach et al., 2020), the Global Forcasting System (GFS) from NOAA National Centers for Environmental Prediction (NCEP, 2000) and the Goddard Earth Observing System-Forward Processing (GESO-FP, Molod et al., 2012). Furthermore, we include the in-situ wind speed measured at Bornholm airport, which is located about 50 km away from the NS2 leak (IEM, 2023). For September 29[th], we obtain wind speeds of 4.1, 6.6, 4.8 and 3.6 m/s from ERA5, GFS, GEOS-FP and airport measurements, respectively; and for September 30[th], we obtain wind speeds of 5.0, 6.3, 6.3 and 5.7 m/s, respectively. **We randomly pick one of these four wind speeds.** |
|---|---|

This 50% wind speed uncertainty was a conservative assumption, following Schuit et al. (2023). To better evaluate the actual uncertainty, we now compare ERA5, GEOS-FP and GFS products with wind speed measured bi-hourly at Bornholm airport for 2022. Over the three reanalysis products, we obtain an averaged standard deviation for "re-analysis – Bornholm airport" differences of 1.6 m/s. This amounts to 34% and 28% of the averaged 10m wind speed obtained from all these data sources at the time of L8 and S-2B overpasses, respectively. Overall, the first-order sensitivity index calculation that we provide show that this wind speed uncertainty contributes little to the emission result variance compared to the contribution of the empirical calibration of sea foam albedo spectral dependence uncertainty. We adjusted the revised manuscript.

| New text: line 228 – 230 | **(5) To account for wind speed error, we evaluate the differences between the three re-analysis models (ERA5, GEOS-FP, GFS) and in-situ measurements made at Bornholm airport for 2022. On average, we find a standard deviation of 1.6 m/s. We therefore sample the wind speed error from a Gaussian distribution with a 1.6 m/s standard deviation and centred on zero.** |
|---|---|

L198: I cannot reproduce the numbers.
In the original version, the ensemble description had one slight imprecision: it lacked to specify that the perturbation on effective wind speed coefficients was applied at the same time on both coefficients, thus possibly causing trouble when trying to reproduce the numbers. In the revised manuscript, with the Monte Carlo ensemble approach, we randomly draw 1000000 members for each satellite, thus fixing this ambiguity.

L218: Re: "1M", is this 1 million? If so, please avoid the shorthand and I don't see why the random draws did not come from ~5 million members, but it might not matters. I simply need to know how the one million members were selected by the authors in order to assess whether this is a biased sample.

"1M" stood for 1 million indeed in the original version. We removed this ambiguous shorthand from the revised manuscript. Because we did not have the same number of ensemble members for L8 and S-2B (because of different minimum albedo interval lengths), we randomly drew (with replacement) 1000000 members from their respective ensembles to get the statistics of the L8 and S-2B average. Because results would change very slightly over different draws of 1000000 members, we reported the mean statistics of L8 and S-2B averages over 100 draws of 1000000 members.

We agree this was a cumbersome way to proceed. It is fixed in the revised manuscript, with the Monte Carlo ensemble approach, we now evaluate the statistics of L8 and S-2B average by combining our 1000000-member single satellite ensembles.

L236 (and in the abstract): hypotheses-> assumptions
We corrected this choice of word in the revised manuscript.

| New text: line 17 | This very specific NS2 observation case challenges some of MBSP and IME implicit **assumptions**, and thus calls for customized calibrations: |
|---|---|

| New text: line 63 | This work first aims to show how Landsat 8 and Sentinel-2B observations of the Nord Stream 2 leak challenge implicit **assumptions** in methods usually applied for Earth-imager methane plume analysis and emission rate quantification. |
|---|---|
| New text: line 276 | We have shown how the unusual observations of a sea foam patch surrounded by dark still sea (and clouds for L8) challenge implicit underlying **assumptions** in both the Multi-Band Single-Pass (MBSP) and Integrated Mass Enhancement (IME) methods. |
| New text: line 287 | Overall, we see our work as a methodological cautionary tale illustrating how implicit method **assumptions** need to be considered and compensated for in unusual observation cases such as this one. |

**References**

Varon, D. J., Jervis, D., McKeever, J., Spence, I., Gains, D., and Jacob, D. J.: High-frequency monitoring of anomalous methane point sources with multispectral Sentinel-2 satellite observations, Atmospheric Measurement Techniques, 14, 2771–2785, https://doi.org/10.5194/amt-14-2771-2021, 2021

Varon, D. J., Jacob, D. J., McKeever, J., Jervis, D., Durak, B. O. A., Xia, Y., and Huang, Y.: Quantifying methane point sources from fine-scale satellite observations of atmospheric methane plumes, Atmospheric Measurement Techniques, 11, 5673–5686, https://doi.org/10.5194/amt-11-5673-2018, 2018.

Jia, M., Li, F., Zhang, Y., Wu, M., Li, Y., Feng, S., Wang, H., Chen, H., Ju, W., Lin, J., Cai, J., Zhang, Y., and Jiang, F.: The Nord Stream pipeline gas leaks released approximately 220,000 tonnes of methane into the atmosphere, Environmental Science and Ecotechnology, 12, 100 210, https://doi.org/https://doi.org/10.1016/j.ese.2022.100210, 2022.

Shine, K. P., Campargue, A., Mondelain, D., McPheat, R. A., Ptashnik, I. V., & Weidmann, D. (2016). The water vapour continuum in near-infrared windows–Current understanding and prospects for its inclusion in spectroscopic databases. Journal of Molecular Spectroscopy, 327, 193-208.

Zhang, Hankui K., et al. "Characterization of Sentinel-2A and Landsat-8 top of atmosphere, surface, and nadir BRDF adjusted reflectance and NDVI differences." Remote sensing of environment 215 (2018): 482-494.

Buchwitz, M., Schneising, O., Reuter, M., Heymann, J., Krautwurst, S., Bovensmann, H., Burrows, J. P., Boesch, H., Parker, R. J., Somkuti, P., Detmers, R. G., Hasekamp, O. P., Aben, I., Butz, A., Frankenberg, C., and Turner, A. J.: Satellite-derived methane hotspot emission estimates using a fast data-driven method, Atmos. Chem. Phys., 17, 5751–5774, https://doi.org/10.5194/acp-17-5751-2017, 2017.

Schuit, B. J., Maasakkers, J. D., Bijl, P., Mahapatra, G., van den Berg, A.-W., Pandey, S., Lorente, A., Borsdorff, T., Houweling, S., Varon, D. J., McKeever, J., Jervis, D., Girard, M., Irakulis-Loitxate, I., Gorroño, J., Guanter, L., Cusworth, D. H., and Aben, I.: Automated detection and monitoring of methane super-emitters using satellite data, Atmos. Chem. Phys., 23, 9071–9098, https://doi.org/10.5194/acp-23-9071-2023, 2023.